# Characterization of Two Glycoside Hydrolases of Family GH13 and GH57, Present in a Polysaccharide Utilization Locus (PUL) of *Pontibacter* sp. SGAir0037

**DOI:** 10.3390/molecules29122788

**Published:** 2024-06-12

**Authors:** Hilda Hubertha Maria Bax, Edita Jurak

**Affiliations:** Bioproduct Engineering, Engineering and Technology Institute Groningen, University of Groningen, Nijenborgh 4, 9747 AG Groningen, The Netherlands; h.h.m.bax@rug.nl

**Keywords:** glycogen branching enzyme, alpha-1,4-transglycosylation, glycogen synthesis, polysaccharide utilization loci, maltooctadecaose, glycogen branching enzymes (EC 2.4.1.18)

## Abstract

Glycogen, an α-glucan polymer serving as an energy storage compound in microorganisms, is synthesized through distinct pathways (GlgC-GlgA or GlgE pathway). Both pathways involve multiple enzymes, with a shared glycogen branching enzyme (GBE). GBEs play a pivotal role in establishing α-1,6-linkages within the glycogen structure. GBEs are also used for starch modification. Understanding how these enzymes work is interesting for both glycogen synthesis in microorganisms, as well as novel applications for starch modification. This study focuses on a putative enzyme GH13_9 GBE (PoGBE13), present in a polysaccharide utilization locus (PUL) of *Pontibacter* sp. SGAir0037, and related to the GlgE glycogen synthesis pathway. While the PUL of *Pontibacter* sp. SGAir0037 contains glycogen-degrading enzymes, the branching enzyme (PoGBE13) was also found due to genetic closeness. Characterization revealed that PoGBE13 functions as a typical branching enzyme, exhibiting a relatively high branching over non-branching (hydrolysis and α-1,4-transferase activity) ratio on linear maltooctadecaose (3.0 ± 0.4). Besides the GH13_9 GBE, a GH57 (PoGH57) enzyme was selected for characterization from the same PUL due to its undefined function. The combined action of both GH13 and GH57 enzymes suggested 4-α-glucanotransferase activity for PoGH57. The characterization of these unique enzymes related to a GlgE glycogen synthesis pathway provides a more profound understanding of their interactions and synergistic roles in glycogen synthesis and are potential enzymes for use in starch modification processes. Due to the structural similarity between glycogen and starch, PoGBE13 can potentially be used for starch modification with different applications, for example, in functional food ingredients.

## 1. Introduction

An important energy storage mechanism of microorganisms is synthesizing glycogen with the use of various enzymes. Glycogen is an α-glucan polymer, synthesized as a response to nutrient limitation and the presence of a carbon source [1,2]. Glycogen consists of glucose residues linked through α-1,4- and α-1,6-linkages. The in vivo synthesis of glycogen encompasses two distinct anabolic pathways: the classical GlgC-GlgA pathway and the novel GlgE pathway [3,4,5,6,7,8]. The classical GlgC-GlgA pathway is widely present in diverse biological domains, spanning bacteria, archaea, fungi, yeast, and animals [9,10]. The GlgC-GlgA pathway involves several enzymes. Initially, extracellular glucose undergoes conversion to glucose-6-phosphate through phosphotransferase activity. Subsequently, phosphoglucomutase (PGM) transforms glucose-6-phosphate into glucose-1-phosphate. ADP- or UDP-glucose is then generated from glucose-1-phosphate by the action of ADP-glucose pyrophosphorylase (GlgC; EC 2.7.7.27) or glucose-1-phosphate uridylyltransferase (GalU; EC 2.7.7.9) in the presence of ATP and Mg^2+^ [3,11]. The resulting ADP/UDP-glucose is subsequently transferred to the non-reducing end of a growing α-1,4-linked glucan chain, leading to the release of ADP/UDP. Glycogen synthase (GlgA; EC 2.4.1.11) plays a crucial role by forming the primer necessary for elongation and extending the α-1,4-chain [3,11,12,13,14]. The introduction of α-1,6-linkages is arranged by glycogen branching enzymes (GBEs; EC 2.4.1.18) [3,5,11]. GBEs represent crucial enzymes in the formation of α-1,6-linkages within the linear chains of glycogen.

The recently elucidated GlgE pathway involves a sequence of enzymatic reactions mediated by four enzymes [15,16]. The process starts with trehalose synthase (TreS; EC 5.4.99.16) and maltose kinase (Pep2; EC 2.7.1.175), forming a complex that catalyzes the conversion of trehalose into maltose-1-phosphate (M1P) [17,18]. Following this, maltosyltransferase (GlgE; EC 2.4.99.16) facilitates the elongation of linear α-1,4 glucan chains by transferring the maltosyl group from M1P to an existing α-1,4 glucan chain [19,20]. This process leads to the extension of the glucan structure. In the context of the GlgE pathway, GBEs play a similar role as its counterpart in the classical GlgC-GlgA pathway, introducing branches into the glucan chain.

Interestingly, the combination of these enzymes involved in the GlgE pathway was found in a polysaccharide utilization locus (PUL) from *Pontibacter* sp. SGAir0037 [21,22], (www.cazy.org/PULDB/, accessed on 1 June 2024). PULs are genomic clusters housing genes responsible for encoding degradation-related carbohydrate-active enzymes (CAZymes), carbohydrate-binding modules (CBMs), carbohydrate transporters, and proteins of unknown function (PUFs). These genetic clusters undergo co-regulation, inducing synchronous upregulation upon exposure to specific substrates. This arranged expression facilitates cooperative activity on the targeted substrate. PULs are predominantly situated within the abundant Bacteroidetes phylum. Organisms belonging to this phylum possess extensive assortments of PULs. This genetic reservoir allows Bacteroidetes to effectively degrade multiple different polysaccharides [21,23]. An automated methodology to predict PULs in sequenced Bacteroidetes genomes has been developed and is housed in PULDB [21]. In PULDB, PULs are identified as physically linked genes organized around a starch utilization system (Sus)C/D gene pair. SusC is an inner membrane-spanning transporter, whereas SusD is a sugar-binding protein [21].

Generally, PULs encode enzymes that sense, bind, and efficiently degrade multiple different carbohydrates by working synergistically [24,25,26,27]. The PUL of *Pontibacter* sp. SGAir0037, next to degradation-related CAZymes, includes several enzymes involved in a synthetic pathway as opposed to a degradation pathway. In this study, the genes from the putative GH13_9 and GH57 enzymes from *Pontibacter* sp. SGAir0037, without previously known function or characterization, were selected, overexpressed, purified, and characterized to investigate their activity and their involvement in the glycogen synthesis pathway.

GH13_9 enzymes are a subfamily of GH13, classified as α-1,4-glycogen branching enzymes. Enzymes belonging to the GH57 family can be classified as α-1,4-glycogen branching enzymes, 4-α-glucanotransferases (EC 2.4.1.25), α-amylases (EC 3.2.1.1), pullulanases (EC 3.2.1.41), or cyclomaltodextranses (EC 3.2.1.54) with PoGH57 having no assigned function. Since these enzymes are found in a PUL with other enzymes involved in the GlgE pathway, they are genetically closely related. PoGH57 was previously reported only as a hydrolase without any predicted function and the PoGBE13 enzyme within such a locus is expected to work synergistically in glycogen synthesis. Therefore, this is the first time a branching enzyme has been reported in a polysaccharide utilization locus as well as in genetic proximity to a GH54 (potentially also branching). 

The activity of these two enzymes was investigated using a highly defined linear substrate maltooctadecaose (MD18). This substrate allows for a detailed investigation of the mode of action [28]. The MD18 substrate consists of linear chains of mainly DP18 and a minor quantity of linear chains with DP17. 

This study gives a better understanding of the function of the GH13_9 and GH57 enzymes in the GlgE pathway. It is hypothesized that these enzymes have a synergistic effect and are involved in glycogen synthesis resulting in an α-glucan with a higher branch density. 

## 2. Results and Discussion

### 2.1. Choice of Enzymes and Their Presence in a PUL

The putative GH13_9 and GH57 enzymes from *Pontibacter* sp. SGAir0037 were selected for research, due to their presence in a PUL (predicted PUL18 from PULDB, Figure 1, https://pubmed.ncbi.nlm.nih.gov/29088389/, accessed on 1 June 2024) together with other enzymes involved in the GlgE pathway of glycogen synthesis. As shown in Figure 1, the selected PUL contains a transcriptional regulator (SusR) and the SusC/SusD gene pair (a binding protein and a transporter) used to identify PULs by the database. This PUL contains all enzymes involved in the GlgE pathway of glycogen synthesis. The presence of enzymes involved in glycogen synthesis in operons is not uncommon, which have a similar function as PULs [29]. The first predicted enzymes are alpha-amylases (GH13s), followed by an α-glucosidase (GH97), and a gene for a glycogen branching enzyme (GH13_9) also containing a carbohydrate-binding module 48 (CBM48). Furthermore, GH13_16 is predicted to be a trehalose synthase (TreS), GH13_3 a maltosyl transferase (GlgE), and GH13_10 a malto-oligosyltrehalose trehalohydolase. A GH57 is present, only annotated as a glycoside hydrolase. Using a basic local alignment search tool (BLAST) for this GH57 did not show similar characterized enzymes (February 2024). Lastly, a malto-oligosyltrehalose synthase (GH13_26) and a 4-alpha-glucanotransferase (GH77) are present in the PUL. This PUL is the only one found in the PULDB database with both a GH13_9 and a GH57.

From this PUL, the CBM48-GH13_9 and GH57 were selected, overexpressed, purified, and characterized in this study. The GH13_9 is a predicted α-1,4-glycogen branching enzyme, and in this study further referred to as PoGBE13. Most studies and characterized GBEs, so far, are involved in the classical GlgC-GlgA pathway whereas PoGBE13 is related to the GlgE pathway. Generally, PULs contain sets of enzymes involved in degradation, but PULs are also gene clusters of enzymes that show genetic closeness. Due to the distance from the SusC/D gene pair to the enzymes, they are included in a PUL. The GH13_9 enzyme is in the opposite direction on the genome (Figure 1), and therefore not necessarily expressed simultaneously with the other enzymes. 

The GH57 enzyme (further referred to as PoGH57) is predicted to be a DUF3536 (domain of unknown function)-containing protein when using BLAST. Since enzymes in the GH57 family can exhibit different catalytic functions, it is not known yet which role this enzyme might have in the context of the PUL. It is known from previous research that a number of bacteria have two genes encoding both a GH13 and GH57 GBE [28]. Therefore, it is investigated whether PoGH57 can be classified as a GBE. Xiang et al., 2021 identified two specific fingerprints present in GH57 enzymes with glycogen branching activity [30]. The first is a set of five amino acids with the combination HxHLP, with x being A, S, or T (Figure 2). This set of amino acids was found in the conserved sequence region I (CSR-I) of almost all GBE sequences. In PoGH57, HGHFY is present at the position of the CSR-I of a well-characterized GH57 GBE from *Thermococcus kodakarensis*. The second GBE fingerprint is the combination of six amino acids ELF(Y)GHW present in CSR-IV (Figure 2) [30]. In PoGH57, ETYGHH is present at the position of the CSR-IV in the GH57 GBE from *T. kodakarensis*. Neither of these two fingerprints (CSR-I and CSR-IV) are present in the amino acid sequence of PoGH57. Therefore, based on the amino acid sequence, this enzyme cannot be described as a GH57 GBE.

#### 2.1.1. Activity of *Pontibacter* sp. SGAir0037 GH13_9 and GH57 Enzymes

To assess the activities of PoGBE13 and PoGH57, following their expression and purification, as confirmed by SDS-PAGE analysis (Appendix A), both were functionally characterized on amylose and linear maltooctadecaose (MD18), demonstrating their roles as functional α-glucan-modifying enzymes.

GBEs catalyze branching through a transglycosylation reaction, utilizing a new α-1,4-glucan chain as an acceptor. A concurrent hydrolytic reaction involves the use of water as an acceptor, resulting in the formation of a shorter α-1,4-glucan chain. In addition to hydrolytic activity, GBEs exhibit elongation activity, creating new α-1,4-linkages and thereby extending the α-glucan chain [31,32]. The branching and non-branching activities of PoGBE13 with amylose and MD18 substrates were monitored over time, using a reducing end assay. Table 1 presents data indicating branching activity on both substrates, although on amylose a 50-fold higher branching activity compared to MD18 (591.4 and 18.0 mU^B^/mg E, respectively) was observed. The ratio of branching to non-branching activity for PoGBE13 on MD18 was determined to be 3.0 ± 0.4, a value comparable to the ratio observed for the GH13 glycogen branching enzyme from *Petrotoga mobilis* (PmGBE13; 3.7 ± 0.3) [28].

The total activity, branching and non-branching, of PoGH57 was also tested on both substrates. PoGH57 showed a low activity on both amylose (27.7 ± 0.7 mU/mg E) and MD18 (4.2 ± 0.2 mU/mg E). The total activity on amylose was 6.5 times higher than on MD18, whereas the activity on amylose for PoGBE13 was 50 times higher than on MD18. PoGH57 shows a low affinity towards amylose compared to PoGBE13.

#### 2.1.2. Optimal Conditions of Pontibacter GH13 and GH57 Enzymes

The impact of temperature and pH on the enzymatic activity of PoGBE13 and PoGH57 was systematically explored. Both enzymes exhibited their highest activity at 30 °C and a pH of 7.0 (Figure 3). However, it was observed that the activity of both enzymes significantly declined at temperatures exceeding 40 °C. The optimal activity at a neutral pH aligns with the characteristics of other GBEs [33]. These optimal growth conditions are in good correlation with the environment where *Pontibacter* sp. SGAir0037 was found. This bacterial strain was isolated from an aerosol sample collected in Singapore (1.346° N, 103.680° E) [34]. 

### 2.2. Chain Length Distribution of Modified Maltodextrin DP18

The linear maltodextrin MD18 was employed as a substrate to enable a comprehensive analysis of changes in chain length distribution. The chain length distribution of MD18 was followed after separate modifications with PoGBE13 and PoGH57, as well as concurrent modifications involving both enzymes (Appendix A). This experimental approach allowed for a detailed investigation of the impact of individual and simultaneous enzymatic modifications on the distribution of chain lengths within the maltodextrin substrate.

Upon incubation of MD18 to PoGBE13, nearly all DP18 molecules were converted within a 24 h timeframe, resulting in the generation of both shorter linear chains and branched chains (Figure 4). The linear chains predominantly fell within the DP7 to DP9 range, while the branches exhibited a broader distribution spanning DP6 to DP10. This is in correlation with previous research on other GH13_9 GBEs. Comparative research has shown that treatment with a *Petrotoga mobilis* GBE13_9 led to the formation of relatively longer linear chains (DP8 to DP12) and shorter branches (DP5 to DP8) [28]. The varying lengths of linear and branched chains between PmGBE13 and PoGBE13 could be related to a different final glycogen structure. As both *Petrotoga mobilis* and *Pontibacter* sp. SGAir0037 live in a different environment, they might also require a different glycogen structure as an energy storage molecule.

In a one-pot reaction involving both PoGBE13 and PoGH57, the chain length distribution pattern after 24 h closely resembled that obtained from the sole action of PoGBE13. However, the number of linear chains is higher, and the number of branches produced is lower in the presence of PoGH57, indicating its possible 4-α-glucanotransferase activity. 

### 2.3. Two-Step Modification of Linear Maltodextrin DP18 with GH13 and GH57 Glycogen Branching Enzymes

Since PoGBE13 and PoGH57 are both in the same PUL, these enzymes are also expected to be expressed simultaneously. To gain a better understanding of the possible synergistic effect of these enzymes on MD18, the impact of a two-step modification on the chain length distribution was tested in an additional experiment. 

Figure 5 presents the chain length distribution of linear and branched chains after a two-step reaction, involving PoGBE13 first followed by PoGH57. When the substrate underwent a 24 h modification with PoGBE13 as the first step, followed by PoGH57 in the second step, it resulted in significantly more short linear chains of DP3 and DP4. The number of short branches of DP3 and DP4 also increased, whereas the number of branches with a length above DP6 decreased. Therefore, the hydrolytic activity of PoGH57 was dominant on this substrate, mainly trimming the branches with a DP higher than six to shorter branches. This is in correspondence with the GH57 GBE from *P. mobilis* previously studied. In a two-step modification with PmGBE13 and PmGBE57, the branch length significantly decreased [28].

The same reaction was also performed with the enzymes in a different order. Figure 6 shows the chain length distribution after an initial modification step with PoGH57, followed by PoGBE13. The first step with PoGH57 shows its hydrolytic activity since there are almost no differences in chain lengths after debranching. In the secondary step with PoGBE13, the remaining MD18 substrate was used to create branches of DP5 to DP8. PoGBE13 was, however, not able to utilize the shorter chains. Therefore, the effect of the branching activity on the hydrolyzed substrate was only minor. This is in correlation with a two-step reaction with PmGBE57 first and PmGBE13 second [28]. 

The total peak area of branched chains, as shown in Figure 7, is highest for treatment with only PoGBE13, and decreases when both enzymes are used in combination. This indicates that PoGH57 only showed hydrolytic activity with this substrate. The α-glucan structure thus gets smaller with shorter branches, but the number of branches stays constant. In contrast, a combination of PmGBE13 and PmGBE57 in a two-step reaction with GBE13 first followed by GBE57 did result in a higher total peak area for the branches [28]. Therefore, should both enzymes work simultaneously, they would have a synergistic effect where the GH13_9 enzyme acts as a branching enzyme and the GH57 as a 4-α-glucanotransferase to create an α-glucan with shorter branches and a denser structure. Comparing the enzymes from *P. mobilis* and *Pontibacter* sp. SGAir0037, it can be speculated that the GH13 and GH57 GBE from *P. mobilis* work together synergistically, resulting in a different α-glucan as the end product. *P. mobilis* was isolated from a North Sea oil-production well [35], whereas *Pontibacter* sp. SGAir0037 was isolated from an outdoor air sample collected in Singapore [34]. Both organisms thus have a different habitat, which might be related to a different glycogen structure.

In *Pontibacter* sp. SGAir0037, the combination of the enzymes also results in a product with shorter branches, but in the combination of these enzymes, PoGH57 acts as a hydrolytic enzyme. It can be concluded that the PUL of *Pontibacter* sp. SGAir0037 contains only one GBE (in family GH13_9), and the GH57 enzyme acts as an α-1,4-glucanotransferase. This observation might be related to the annotation of these enzymes as a PUL since typically PULs contain enzymes that degrade specific substrates.

## 3. Materials and Methods

### 3.1. Materials

The linear maltodextrin MD18 (Maltooctadecaose) was obtained from CarboExpert (Daejeon, Yuseong-gu, Republic of Korea), while linear maltodextrin MD7 (Maltoheptaose) was attained from Sigma-Aldrich (Saint Louis, MO, USA). Isoamylase from *Pseudomonas* sp. (E-ISAMY, 200 U/mL) and pullulanase M1 from Klebsiella planticola (E-PULKP, 650 U/mL) were sourced from Megazyme (Bray, Ireland). MagicMedia and HisPurTM Ni-NTA Resin were acquired from ThermoFisher Scientific (Waltham, MA, USA). All utilized chemicals were of analytical grade or higher quality.

### 3.2. Enzyme Production and Purification

The genes encoding the GH13_9 glycogen branching enzyme from *Pontibacter* sp. SGAir0037 (PoGBE13, Genbank: QCR22793.1) and the GH57 enzyme from *Pontibacter* sp. SGAir0037 (PoGH57, WP_137759307.1) were codon-optimized by GenScript, cloned into a pET28a(+) vector containing a C-terminal His-Tag for purification (GenScript USA Inc., Piscataway, NY, USA), and expressed in *E. coli* BL21(DE3). The enzymes were overexpressed and purified with a HisPur^TM^ Ni-NTA column, as described in Bax et al., 2023 [28].

The iodine assay was employed to evaluate the activity of potato amylose V (AVEBE, Groningen, The Netherlands) [36]. The enzymes were incubated at different concentrations (25–300 µg/mL) with 1 mg/mL potato amylose in a 50 mM sodium phosphate buffer, pH 7.5 at 50 °C. At various time points (10 min to 2 h), 15 µL aliquots were withdrawn from the enzyme reactions and mixed with 100 µL of a freshly prepared iodine reagent (0.26% KI, 0.026% I2, 5 mM HCl). After the final aliquot, the absorbance of the iodine–amylose complex was measured at 610 nm using a spectrophotometer (SpectraMax Plus 384 Microplate Reader, Molecular Devices, Sunnyvale, CA, USA). One unit of activity is defined as the decrease in absorbance of 1.0 per min at 610 nm.

The iodine assay was also used to determine the optimal pH and temperature of PoGBE13 and PoGH57. The influence of temperature and pH on the activity of both enzymes was tested in 50 mM sodium phosphate buffer at pH 7.0 or RT, respectively. The temperature ranged from 20 to 60 °C and the pH from 5.0 to 10.0. For pH 5.0, a 50 mM sodium acetate buffer was used, for pH values in the range of 6.0 to 8.0, a 50 mM sodium phosphate buffer was used, and for pH 9.0 and 10.0, a 50 mM Tris buffer was used. Amylose at a concentration of 1 mg/mL was used as substrate.

### 3.3. Enzyme Reactions and Analysis of Activity with Reducing End Assay

The determination of branching and non-branching activity of PoGBE13 and PoGH57 involved incubating the enzymes with 2.5 mg/mL potato amylose in 10 mM sodium phosphate buffer (pH 7.5), slowly rotating head-over-tail at 50 °C. The enzyme concentrations at which the initial reaction resulted in a linear increase in activity were determined to be 1.5 and 20 mg/g substrate for PoGBE13 and PoGH57, respectively. Reactions were carried out for 0.5, 1, 1.5, 2, and 2.5 h, and terminated by boiling for 5 min. For the debranching reaction, the GBE-modified glucans were diluted in a sodium acetate buffer, pH 4.5, and treated with isoamylase (1 U/mg substrate) and pullulanase (0.7 U/mg substrate) for 24 h at 40 °C. 

An analysis of branched and debranched samples was performed using the pAHBAH-reducing end assay, as described in Bax et al., 2023 [28]. One unit of activity is defined as 1 µmol reducing ends released or transferred per minute.

As previously introduced, the definition of activity U^B^ (Units-Branching activity) facilitates a fair comparison of branching activity by excluding products from hydrolytic and elongation activity [28,32]. The increase in reducing ends from branched to debranched products represents the branching activity, U^B^, and is defined as follows:$$U^{B}=\frac{\Delta RE\;after\;debranching\;\left[\mu mol\right]-\Delta RE\;before\;debranching\;[\mu mol]}{\Delta time\;[min]}$$

One U^B^ of activity is defined as 1 μmol reducing ends transferred to a branching point per minute.

The branching and debranching reactions were repeated with MD18 as a substrate (0.5 mg/mL), with an enzyme dose of 0.5 U^B^/g substrate (units, U^B^, based on branching activity on amylose).

### 3.4. Chain Length Distribution with Anion Exchange Chromatography

To investigate and compare the mode of action of PoGBE13 and PoGH57, an incubation was conducted with MD18 (0.5 mg/mL) and an enzyme dose of 1 U^B^/g substrate for all enzymes (units, U^B^, based on branching activity on MD18). Both branched and debranched samples of this incubation were subjected to analysis using High-Performance Anion Exchange Chromatography coupled with Pulsed Amperometric-EDet1 Detection (HPAEC-PAD) with the Gold Standard PAD waveform. This analysis was performed using a Dionex ICS-6000 system (ThermoFischer Scientific, Waltham, MA, USA) equipped with a CarboPacTM PA100 column. The chain length distribution analysis was performed as previously described by Bax et al., 2023 [28]. Peak areas were calculated using the Chromeleon software version 7.2.9. The increase in linear chains was calculated as Δ peak area, representing the difference in peak area from the untreated substrate to 24 h modification, and the Δ peak area of branched chains was calculated by the increase in peak area after the debranching of 24 h modified samples minus the increase in peak area after the debranching of the untreated substrate.

### 3.5. Statistical Analysis

Data are presented as means of triplicates with standard deviations. The statistical analysis was conducted in Stata17 (StataCorp, College Station, TX, USA) using a one-way analysis of variance (ANOVA) with a post-hoc Bonferroni test and a 95% confidence interval.

## 4. Conclusions

Two unique enzymes from a polysaccharide utilization locus (PUL) of *Pontibacter* sp. SGAir0037, namely PoGBE13 and PoGH57, were successfully overexpressed, purified, and characterized. Both enzymes exhibited activity on amylose and MD18. PoGBE13 demonstrated higher total activity compared to PoGH57, along with a high branching versus non-branching ratio. This pattern aligns with observations from previous studies on GH13 GBEs [28,30,31,32,33].

The combined action of both enzymes indicated that PoGH57 predominantly played a hydrolytic role, while PoGBE13 was responsible for creating branches. In the context of the two-enzyme combination on linear maltodextrin, PoGH57 showed α-1,4-glucanotransferase activity. The presence of one GBE and one α-1,4-glucanotransferase relates to the presence of these enzymes in a PUL with other enzymes related to the GlgE pathway for glycogen synthesis. The interaction of PoGBE13 and PoGH57 with other enzymes in the PUL might alter its role in the synthetic pathway. Further research on these unique enzymes, as well as the other enzymes within the PUL, holds the potential to provide a deeper understanding of their interactions and synergistic roles in the GlgE pathway of glycogen synthesis.

## Figures and Tables

**Figure 1 molecules-29-02788-f001:**
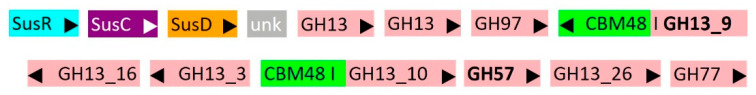
The predicted PUL18 from *Pontibacter* sp. SGAir0037 (PULDB, http://www.cazy.org/PULDB/, accessed on 1 June 2024).

**Figure 2 molecules-29-02788-f002:**
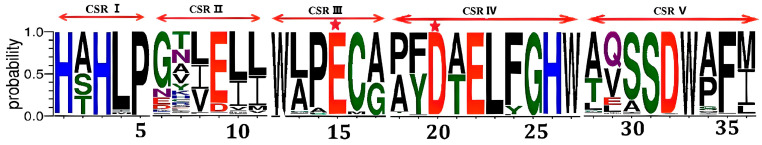
Fingerprint logos of five CSRs among 2447 glycoside hydrolase family 57 glycogen branching enzyme homologues; the two catalytic residues are indicated by a red star symbol (*). Adapted from Xiang et al., 2021 [30].

**Figure 3 molecules-29-02788-f003:**
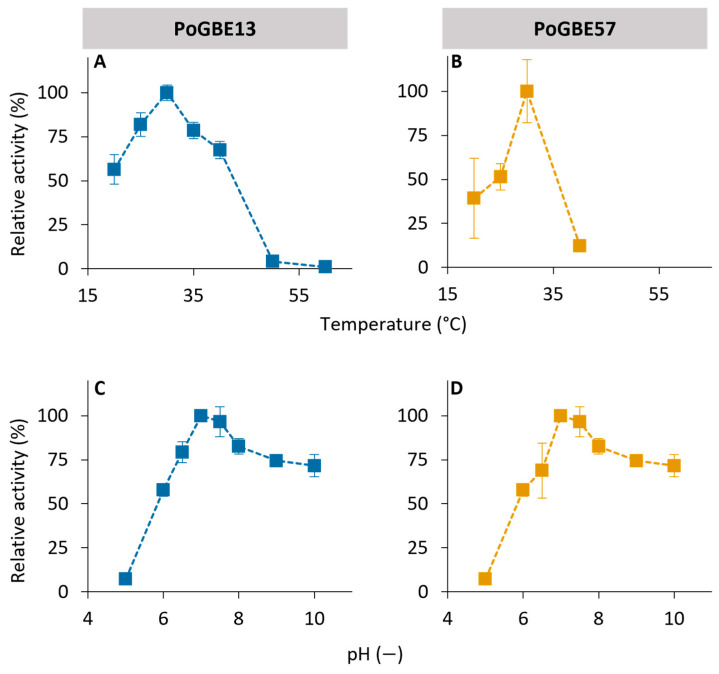
Temperature and pH activity profiles of PoGBE13 (**A**,**C**) and PoGH57 (**B**,**D**).

**Figure 4 molecules-29-02788-f004:**
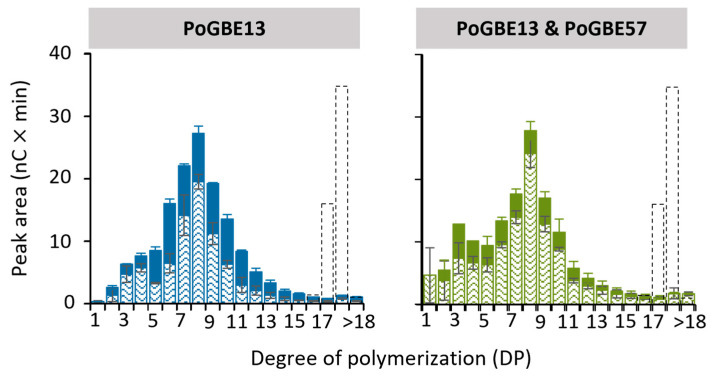
Peak areas of MD18 in linear (pattern fill) and branched (solid fill) chains after treatment with PoGBE13 or PoGBE13 and PoGBE57 at 1 U^B^/g S for 24 h compared to the untreated substrate (dotted bars). The peak area of branched chains is calculated by the increase in peak area after debranching of 24 h GBE-modified samples minus the peak area of the 24 h GBE-modified samples before debranching.

**Figure 5 molecules-29-02788-f005:**
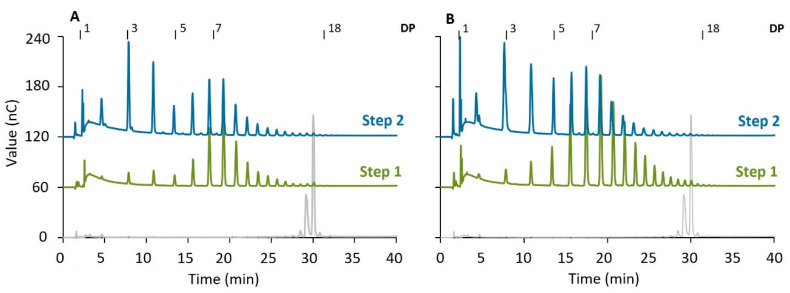
The chain length distribution of MD18 modified with PoGBE13 (step 1) and PoGBE57 (step 2) at 1 U^B^/g S in a two-step modification for 24 h before (**A**) and after (**B**) debranching compared to the untreated substrate (grey).

**Figure 6 molecules-29-02788-f006:**
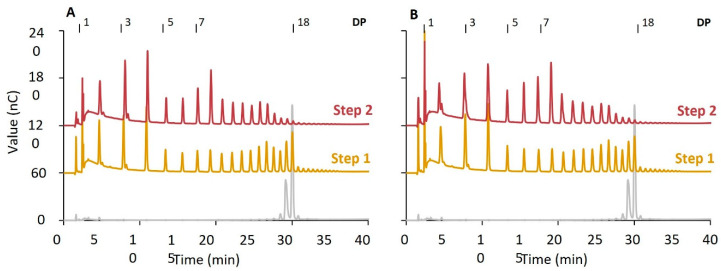
The chain length distribution of MD18 modified with PoGBE57 (step 1) and PoGBE13 (step 2) at 1 U^B^/g S in a two-step modification for 24 h before (**A**) and after (**B**) debranching compared to the untreated substrate (grey).

**Figure 7 molecules-29-02788-f007:**
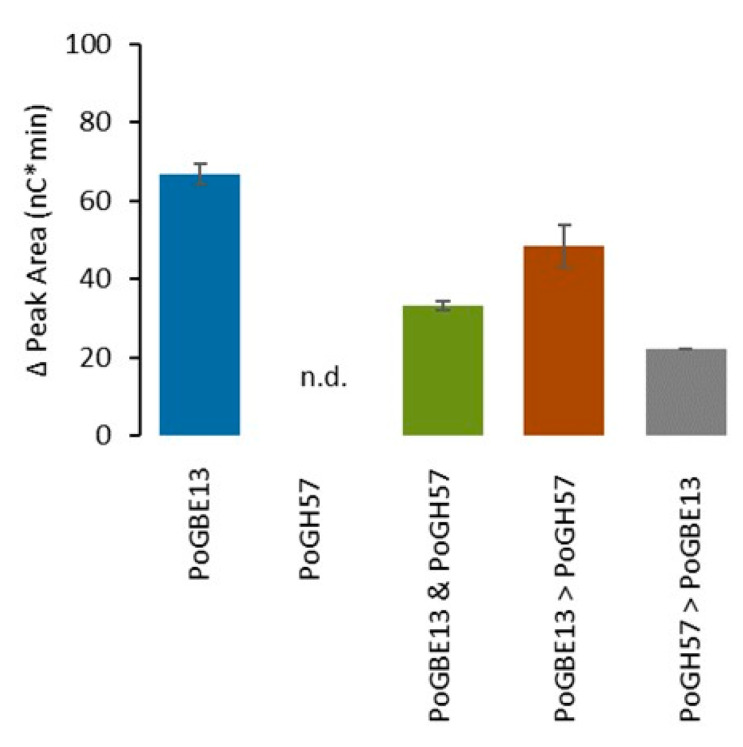
Total increase in peak area of branched chains after treatment of MD18 with PoGBE13 or PoGBE13 and PoGBE57 in one or two steps at 1 U^B^/g S for 24 h. n.d.: the increase in peak area of only PoGH57 was not detectable, since PoGH57 has almost no branching activity.

**Table 1 molecules-29-02788-t001:** Activity of PoGBE13 (*Pontibacter* sp. SGAir0037 glycogen branching enzyme GH13) on amylose and MD18, analyzed as increase (non-branching activity) or transfer (branching activity) of reducing ends.

	PoGBE13	
	Amylose	MD18
Non-branching activity [mU^NB^/mg E]	11.8 ± 3.1	6.1 ± 0.9
Branching activity [mU^B^/mg E]	591.4 ± 71.6	18.0 ± 0.7
Ratio B:NB *	51.9 ± 7.3	3.0 ± 0.4

± average of three independent measurements with standard deviation. * ratio branching activity to non-branching activity.

## Data Availability

Raw data generated during the experiments described in this article are available upon request to the corresponding author, Dr. E. Jurak (e.jurak@rug.nl).

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
