# Peer review of "Characterization of Two Glycoside Hydrolases of Family GH13 and GH57, Present in a Polysaccharide Utilization Locus (PUL) of Pontibacter sp. SGAir0037"

_molecules, 2024, doi:10.3390/molecules29122788_

Round 1
Reviewer 1 Report
Comments and Suggestions for Authors
The overall quality of the English language in the manuscript is acceptable; however, minor editing is recommended to enhance clarity and coherence. Some sentences could benefit from a smoother flow and better organization of ideas.
Reviewer 2 Report
Comments and Suggestions for Authors
Abstract written heavy information like introduction.
Abstract need revision as it can expose the key results and application as well as aim .
Line 249 mentioned total activity on amylose was 6.5-times higher than on MD18. For what reason this higher on this only? could you justify.
PoGBE13 350 demonstrated higher total activity compared to PoGH57. Expalin with figure if possible.
what the main aspect adressed to expose from this research?
Results and discussion highlight the key results.
Fig.2. its not understandable. Make it easy for readers understand.
Figures. What on x- axis?
Table.1. what does it mean + or - indicate?. Mention in the table.
Conclusion also need amendments like key results highlight.
Comments on the Quality of English LanguageMinor editing
Reviewer 3 Report
Comments and Suggestions for Authors
This manuscript reports to characterize two glycogen-biosynthesized enzymes (referring to PoGBE13 and PoGH57) from the PUL cluster in the bacteria Pontibacter sp. SGAir0037. With the recombinant enzymes overexpressed in E. coli, PoGBE13 was demonstrated as a branching enzyme exhibiting a relatively high branching over non-branching ratio on linear maltooctadecaose (3.0 ± 0.4). Through a two-step modification of maltooctadecaose, PoGH57 was suggested as a 4-α-glucanotransferase. Conclusively, this study provides us with new insights to understand the GlgE glycogen biosynthesis in bacteria.
Specific comments are also proposed as following:
1. L26-27, I suggest this sentence could be deleted.
2. L102, before the refence [31], [28][29] was not cited.
3. L123, the reference [28] should be corrected as [29],also in L151 and L172.
4. L136, the temperature symbol “°C” should be corrected as “°C”.
5. L187, before the reference [35], not see [32] and [33].
6. L219, the reference [31] should be corrected as [30]
7. L247, what is “the total activity” for PoGH57, branching plus non-branching?
8. In abstract and conclusions, the authors repeatedly mention “unique enzymes” referred to PoGBE13 and PoGH57. What does the “unique” refer to? As mentioned in the manuscript, this pair of PoGBE13 and PoGH57 is somehow similar with the previously described PmGBE13 and PmGBE57.
